# The COVID-19 Pandemic and the Acceptance of E-Learning among University Students: The Role of Precipitating Events

Prageeth Roshan Weerathunga [1,*] , W. H. M. S. Samarathunga [1,*] , H. N. Rathnayake [1], S. B. Agampodi [2], Mohammad Nurunnabi [3] and M. M. S. C. Madhunimasha [1]

1 Faculty of Management Studies, Rajarata University of Sri Lanka, Mihintale 50300, Sri Lanka; hashanr@mgt.rjt.ac.lk (H.N.R.); sanuri1996@gmail.com (M.M.S.C.M.)
2 Department of Community Medicine, Faculty of Medicine and Allied Sciences, Rajarata University of Sri Lanka, Saliyapura 50008, Sri Lanka; suneth.med@rjt.ac.lk
3 Department of Accounting, Prince Sultan University, P.O. Box 66833, Riyadh 11586, Saudi Arabia; mnurunnabi@psu.edu.sa
* Correspondence: prageeth@mgt.rjt.ac.lk (P.R.W.); manoj.thm@mgt.rjt.ac.lk (W.H.M.S.S.); Tel.: +94-71-2483103 (P.R.W.); +94-71-8134115 (W.H.M.S.S.)

**Abstract:** This study examined the effect of the COVID-19 pandemic and related events on the use of e-learning, as well as other key determinants of it. The data were collected from 1039 university students in Sri Lanka. To examine the influence of the COVID-19 pandemic, which was viewed through the lens of precipitating events, on the intention–behaviour relationship, we employed the Technology Acceptance Model (TAM) with the inclusion of a moderating variable. While the findings indicated that the COVID-19 pandemic had clearly increased the usage of e-learning, we found no evidence to establish a moderating impact on the intention–behaviour relationship. The empirical model, however, was well fitted to the data, and the key components of the TAM were likewise adequately described by the relevant predictors. Furthermore, attitudes toward e-learning and perceived ease of use emerged as the most important factors in explaining behavioural intention, whereas relevance and experience were shown to be more significant in relation to perceived usefulness and perceived ease of use. Our work is significant because it adds to the existing empirical evidence on e-learning and supports the relevance of TAM in understanding the usage of e-learning even in extreme situations such as the COVID-19 pandemic. Our research has significant implications for educators and other higher education authorities.

**Keywords:** the COVID-19 pandemic; e-learning; Sri Lanka; TAM; university students

## 1. Introduction

The Coronavirus disease of 2019 (COVID-19) rampaged across the globe, and as a result, half of the world was forced to shut down by April 2020. Countries all across the world have taken extreme measures to contain the spread of the virus. While these measures are indispensable, they have serious repercussions on every aspect of human lives. For example, the temporary closure of schools, universities, and other educational institutions has forced over 91% of students worldwide, about 1.6 billion, to remain indoors, unable to attend their studies as usual [1]. Such a disruption in the education sector is unprecedented. Moreover, the impact of this crisis on the education of the most vulnerable and marginalised students is enormous. However, despite the fact that the COVID-19 pandemic is disrupting worldwide learning and education, governments and various organisations are striving to provide their students with a "non-stop teaching and learning" experience as much as possible [2]. Soon after the outbreak of the COVID-19 in Wuhan, China, millions of Chinese students and teachers migrated from traditional learning to online learning [2]. For instance, Peking University launched live online programs for its 2613 undergraduate courses and 1824 graduate courses facilitating more than 44,700 students who stay at homes or

dormitories [3]. The United Nation Children's Fund (UNICEF), Microsoft Corporation, and Cambridge University announced the expansion of their global online learning platform called "Leaner Passport" to provide education to students affected by the COVID-19 pandemic [4]. While this massive shift in traditional in-class face-to-face education is a huge challenge for both teachers and students, it is unknown how these changes have affected the students learning behaviour and thus the effectiveness of online learning or e-learning. Although extant studies have emphasised certain obstacles to e-learning, such as lack of self-discipline, motivation, appropriate study materials, and a good learning atmosphere [2], identifying how present changes in the learning environment (due to the COVID-19 outbreak) have influenced student learning behaviour—in particular, their e-learning behaviour—is timely and important.

E-learning, which is defined as "an instructional process that gives online learners access to a wide range of resources—teachers, other learners, and content such as readings and exercises—independently of place and time" [5] (p. 442), has emerged as a result of rapid developments in information technology (IT) over the past decade [6]. It has already been widely used in the education sector [7], particularly in higher education. Even before the COVID-19 pandemic, many colleges and universities around the world offered at least some courses online. Online learning has become more popular as it is economically, technologically, and operationally more feasible [8]. Compared with mainstream education, e-learning requires minimal resources, is open to large numbers of students given their geographical dispersion, and can be easily implemented with the ubiquity of IT [8]. In view of these advantages, e-learning has now become increasingly important as the only feasible option for providing education during the COVID-19 pandemic.

A major impediment when it comes to the use of e-learning in education is the varying degree of acceptance among the students. Indeed, the acceptance of e-learning depends on a multitude of factors [9,10]. However, the consensus on which factors cause the actual use of e-learning is far from settled. This has led to the development of numerous technology adoption theories over time. These theories explain the interrelationship between certain factors that determine the acceptance and actual use of e-learning by certain individuals or groups [9]. The Technology Acceptance Model (TAM) [11], among others, is the most widely used theory in the e-learning literature. Along with this theoretical development, abundant research has been carried out to identify factors that determine the acceptance of e-learning [9,10]. Although previous studies provide solid evidence on the various factors that determine the use of e-learning [9,10], most of these studies have been carried out in an ordinary environment where students may experience both traditional and e-learning at the same time (i.e., blended learning). To date, it is unclear how specific circumstances such as crisis and pandemic influence the students' e-learning behaviour. Some recent incidents show that the participation of elementary school students in online learning during the outbreak of COVID-19 is significantly low in some areas of the United States (US) [12]. However, such incidents are inconsistent with the idea that student participation in e-learning during the COVID-19 crisis should be higher than the pre-crisis period, given that e-learning is the only viable option for education. In addition, there is also a lack of documented evidence on the impact of the COVID-19 pandemic on the e-learning behaviour of university students. This is an important issue as there is an unprecedented shift in traditional learning to online learning in universities around the world during the current pandemic. While there are many issues surrounding this movement in education, it is interesting and timely to investigate students' e-learning behaviour and thus identify the factors that support or hinder effective e-learning for students.

Sri Lanka provides an opportune experiment setting for this study, as all 15 state universities have been closed since 14 March 2020, some time after the COVID-19 outbreak in the country. This has led about half a million undergraduate and graduate students in the country to temporarily halt their usual learning. In the meantime, the University Grants Commission (UGC) has taken several measures to provide "non-stop education" to students through online learning systems. Furthermore, policy decisions have been

taken at the higher authorities (i.e., Ministry of Higher Education and UGC) to provide certain facilities to enable effective e-learning during this pandemic. For example, students can now access university-based Learning Management Systems (LMS) free of charge through any internet service provider. Moreover, faculty members are working relentlessly to transform the course contents so that they can deliver the curriculum effectively through virtual classrooms. Many universities in Sri Lanka have been using Moodle, a free and open-source LMS, since long before the COVID-19 epidemic [13]. However, previous studies have shown that the success of e-learning depends on the level of acceptance among university students [13]. On the other hand, several studies have found that current e-learning systems in higher education institutions in developing countries are not successful due to various challenges [14,15]. These issues have been extensively studied in the current literature [9,16–19]. Yet, the question remains unanswered as to how the current global health crisis and its consequences influence the use of e-learning among university students.

Therefore, this study addresses the following research questions:

1. What are the factors that support or hinder the use of e-learning among undergraduates in Sri Lanka?
2. How do the COVID-19 pandemic and its consequences (curfew, lockdown, isolation, stay home, and other social distancing measures) affect the use of e-leaning among undergraduates in Sri Lanka?

Our work provides a substantial contribution to the literature by answering these two questions. First, amid a massive transformation in conventional face-to-face in-class teaching and learning to fully online mode, there is a growing need to assess the crucial components of effective e-learning implementation. The abrupt shift in the learning environment, as well as a dearth of research, has heightened the necessity for studies, particularly in developing nations such as Sri Lanka. Providing up-to-date and precise findings from a relatively large-scale study, we expand present empirical evidence relating to the acceptance of e-learning. Second, this study demonstrated the importance of employing multiple theoretical perspectives in explaining e-learning behaviour during extreme situations. We sought to enhance our understanding of e-learning behaviour during the COVID-19 epidemic by linking the dimensions of TAM with other behavioural theories, such as the Unified Theory of Acceptance and Use of Technology (UTAUT) and the entrepreneurial potential model. This also contributes to the intention–behaviour gap. Third, the integration of a novel construct to explain the influence of the COVID-19 pandemic and related events on e-learning behaviour, which were characterised as precipitating events, adds considerably to the literature. Finally, we provide insightful findings for educators, administrators, and other officials in Sri Lanka's higher education system that may be useful in implementing appropriate measures for boosting e-learning among the university community.

The remainder of this paper is structured as follows. Section 2 contains a review of the literature and the development of hypotheses. Section 3 discusses the research methods and materials. The data analysis and results are provided in Section 4. Section 5 discusses the findings and their implications, followed by concluding remarks in Section 6.

## 2. Literature Review and Hypotheses Development

### 2.1. Theoretical Framework

With increasing adaption of online learning and teaching around the world, a very dynamic educational environment has spawned immense interest among researchers and other stakeholders [8]. Researchers, in particular, have addressed various issues pertaining to e-education and this is reflected in the abundance of research available related to online learning and teaching (see meta-analysis by [9]). The facets of e-learning that have been extensively studied include, among others, acceptance of e-learning [16,19,20], attitudes towards e-learning [16,19,21], behavioural intention and actual usage of e-learning [16,17,22], problem and challenges for implementing e-leaning [8,23], etc. Moreover, numerous

theoretical models have been developed in the domains of psychology, sociology, and information systems to identify the factors leading to the actual usage of e-leaning. Initially, the Theory of Reason Action (TRA) [24] and TAM [11] were widely used in predicting users' behaviour and explaining the e-learning adoption process [25]. However, the advent of the learning management system (LMS) and the virtual learning environment have created a void in empirical research on the e-leaning acceptance model [25]. As a result, different models have been developed and tested in the technology education literature. These include Theory of Planned Behaviour (TPB) [26], extended versions of TAM such as TAM2 [27] and TAM3 [28], and more recently developed acceptance models such as Unified Theory of Acceptance and Use of the Technology (UTAUT) [29] and General Extended Technology Acceptance Model for E-Learning (GETAMEL) [9].

Previous studies have shown that most universities around the world are using e-leaning systems to deliver their curriculum as a part of their blended learning approach [30]. While e-leaning provides many benefits for students and teachers, such benefits can only be realised when students actually use it. For this reason, increasing attention has been paid to uncover critical factors that determine the actual use of e-leaning. Employing various theoretical models (i.e., TRA, TPB, TAM, UTAUT, GETAMEL), researchers have identified various factors that affect the use of e-leaning by different users (i.e., students, employees, teachers) [9]. These factors can be seen as external factors and the core factors of the technology acceptance models. There are many external factors that have been evaluated by scholars. Among them, subjective norms, relevance, self-efficacy, enjoyment, computer anxiety, and facilitating conditions have constantly been shown to have a significant impact on the users' cognition that ultimately leads to the use or non-use of e-learning [9]. Unlike external factors, there are few core factors such as perceived ease of use, perceived usefulness, attitudes, and use intention that are integral parts of most of the acceptance models [9,16,17,25].

The theoretical model developed in this study is derived from the TAM proposed by [11]. The TAM is rooted in TRA with the exclusion of subjective norms and inclusion of two additional variables, perceived usefulness and perceived ease of use, which are more specific to computer usage behaviour [31]. The TAM and its extended versions (i.e., TAM2, TAM3, GETAMEL) have been extensively used in e-learning literature [9,10,16,19,32,33]. Consistent with the e-learning literature, we developed the theoretical model based on TAM and incorporated several external variables. Especially, following the literature related to UTAUT theory, facilitating conditions were introduced to the model, as this could be a decisive factor in determining undergraduates' e-learning usage in developing countries such as Sri Lanka. Furthermore, drawing on entrepreneurial potential model, the effect of the COVID-19 outbreak and its consequences are integrated into the model through precipitating events. Figure 1 depicts the empirical model, in which the core constructs of the model are denoted by rectangles with thick outline and the rest are external variables. Each arrow connecting one construct to another is a hypothesis developed in this study.

*2.2. Hypothesis Development*

2.2.1. Core Predictors

Behavioural Intention

As in many behavioural theories and technology acceptance theories, the intention is a crucial factor, which acts as an immediate predecessor of actual behaviour. For instance, intention to perform a behaviour is a central factor in behavioural theories such as TRA and TPB, and technology acceptance theories such as TAM, UTAUT, and GETAMEL. Ajzen [26] suggested that intentions capture the motivational factors that affect a behaviour and indicate how much of an effort people are willing to put in order to perform a given behaviour [26]. According to Ajzen [26] (p. 181) "the stronger the intention to engage in a behaviour, the more likely should be its performance". There is ample evidence of the relationship between intention and actual behaviour (i.e., actual use of e-learning) in the e-learning literature (see the meta-analysis [10,18,19,32,34]). Therefore, we posit that:

**Hypothesis 1 (H1).** *Behavioural intention positively affects actual use of e-learning.*

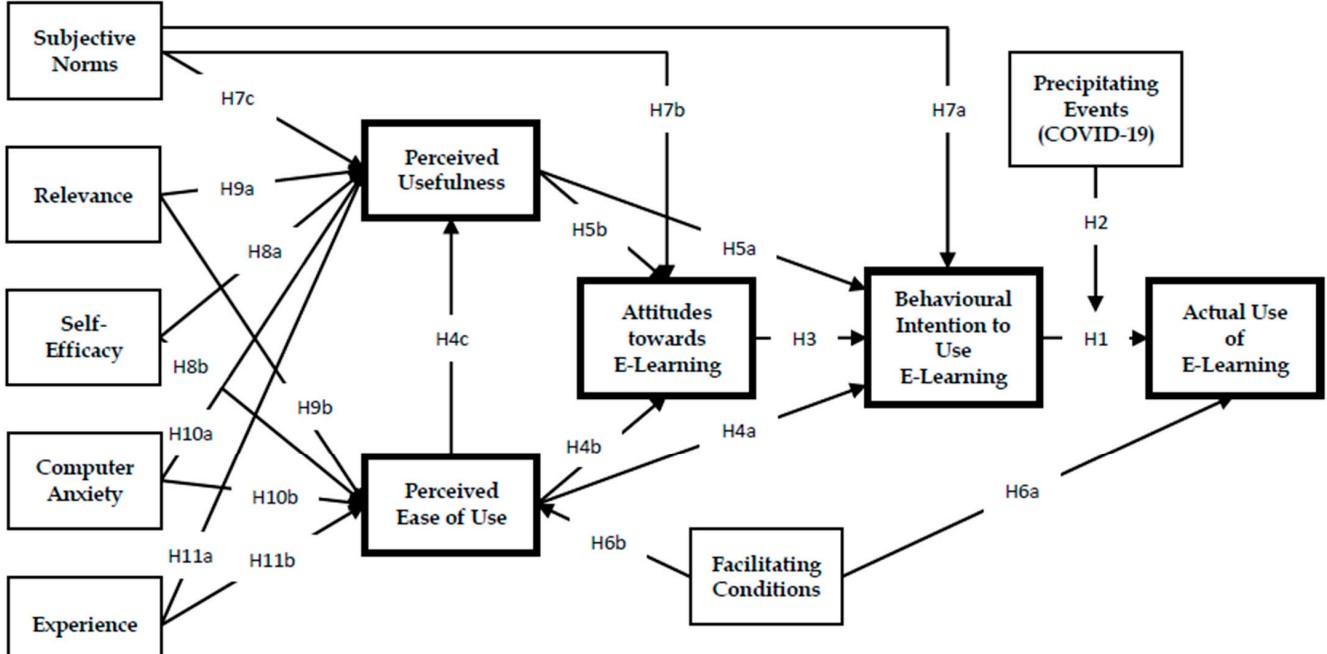

**Figure 1.** Theoretical model for e-learning behaviour.

Precipitating Events (Effect of COVID-19)

Although prior studies have established that the behavioural intention is causal prior to the actual behaviour [35], behavioural intention, however, does not necessarily cause actual behaviour and can only explain less than half of the variance in actual behaviour [10,13,25,35]. Ajzen [26] successfully incorporated perceived behavioural controls into TRA, and proposed an extended version of TRA called the TPB. Perceived behavioural controls in the TPB account for factors that inhibit action such as indispensable resources, competencies, individual or organisational limits, and unforeseen events. These factors are assumed to have direct effect and indirect effects through intention to perform a given behaviour on the realisation of the actual behaviour [26]. However, the intention–behaviour gap has remained unfilled and gained increasing attention from scholars [35].

In an effort to bridge the intention–behaviour gap in the context of technology acceptance, Venkatesh et al. [29] developed the UTAUT model incorporating perceived behavioural controls, which were taken from the TPB, and facilitating conditions. They argued that the facilitating conditions could reflect users' perception on their control over behaviour. However, the same argument was invalidated by Venkatesh [28], pointing out that facilitating conditions could not be a good predictor of behaviour in situations where individuals have insufficient and uncertain information about the behaviour in question.

Additionally, scholars criticise that none of these variables have successfully captured the influence of external factors on the performance of behaviour [35]. Thus, we can observe various attempts in the literature to capture the effect of external factors on the intention–behaviour relationship employing different variables. Moghavvemi et al. [35], for example, used precipitating events as a moderating variable to capture the effect of external factors on the use behaviour of IT innovation. They argued that certain unforeseen events could facilitate or hinder entrepreneurs' intention to use IT innovation and affect the use behaviour. Taking a sample of 351 SMEs in Malaysia, Moghavvemi et al. [35] investigated if and how certain external events such as changes in government policy, resource availability, being offered a big contract, access to a new market, resource availability, etc., change entrepreneurs' use behaviour of IT innovation. The findings of their study showed that

these external events, conceptualised as precipitating events, could trigger the intention to use IT innovation towards actual use behaviour.

According to Krueger [36], precipitating events can be defined as certain changes in the external environment or in other words external factors that could accelerate or "precipitate" the actualisation of behavioural intention into behaviour. The unexpected changes in the environment can lead to abrupt changes in the behaviour of affected individuals. For instance, Shapero and Sokol [37] asserted that significant life events such as sudden unemployment and displacement could cause a considerable increase in entrepreneurial activity. Likewise, Moghavvemi et al. [35] showed that changes in entrepreneurs' daily work environment such as increase in cost, new investment, availability of resources, and customer or supplier request greatly impact the realisation of use intention into use behaviour of IT innovations in their businesses.

Following these arguments in the Information Systems (IS) and entrepreneurship literature, we posit that certain occurrences (i.e., stay home, curfew, lockdown and other travel restrictions) following the COVID-19 pandemic may have accelerated students' behaviour towards more use of e-learning. There are several contributing factors for students' behavioural changes in e-learning during the COVID-19 pandemic, including the massive migration from traditional learning to e-learning in the entire education system, certain improvements in facilitating conditions such as free access to certain web-based resources and the introduction of various data packages at discounted prices that allow them to surf the internet, and more time to spend on computer or other electronic devices due to stay-home orders and other travel restrictions. Additionally, university students had been using e-learning to a certain extent even before the pandemic and hence, the intention to use e-learning may have already formed. When the changes that occurred in the external environment following the COVID-19 pandemic—i.e., stay home orders, travel restrictions, and lockdowns—intervene with the intention–behaviour relationship, the likelihood of realisation of intention to use e-learning into actual usage of e-learning is much greater. Thus, we formulate the following hypothesis:

**Hypothesis 2 (H2).** *The relationship between behavioural intention and the actual use of e-learning is moderated by precipitating events (stay home, isolation, curfew, lockdown caused by COVID-19 pandemic).*

Attitudes towards E-Learning

The relationship between attitudes and behavioural intention has been well studied in behavioural literature. Extant studies view attitude as a significant predictor of intention to perform a behaviour and reflects a person's overall assessment of a behaviour in question [26].

Attitudes regarding a certain behaviour might be positive/negative or favourable/unfavourable [38]. A positive or favourable attitudes is often considered as complementary to the realisation of actual behaviour through intention to perform the behaviour [38]. There are many studies that have shed light on the connection between attitude and e-learning behaviour of students [39–41]. More recent literature also firmly approves the positive association between attitudes and behavioural intention in the context of e-learning [9,16,33]. Therefore, we propose the following hypothesis:

**Hypothesis 3 (H3).** *Attitudes towards e-learning positively affect behavioural intention.*

Perceived Ease Use

Perceived ease of use is a key dimension that affects both behavioural intention and attitudes towards e-learning. Davis defined perceived ease of use as "the extent to which a person considers that the use of a system is free of effort" ([11], p. 320). Accordingly, when users perceived a system to be easy to use, they are more likely to accept it [11]. Previous studies confirmed a significant association between perceived ease of use and

perceived usefulness, attitudes, and behavioural intention in the context of e-learning. For example, Abdullah [9] analysed many scholarly papers on perceived ease of use in e-learning and identified a positive relationship between the perceived ease of use and perceived usefulness. Many researchers hypothesised that perceived ease of use of e-learning increases with the perception of how enjoyable the e-learning system is and the intention to use the system [9,33]. The strong strings between the behavioural intention and ease of use was validated by previous studies including [16,17,28,29]. Therefore, we propose the following hypotheses:

**Hypothesis 4a (H4a).** *Perceived ease of use positively affects behavioural intention.*

**Hypothesis 4b (H4b).** *Perceived ease of use positively affects attitudes towards e-learning.*

**Hypothesis 4c (H4c).** *Perceived ease of use positively affects perceived usefulness of e-learning.*

Perceived Usefulness

Davis ([11], p. 320) defined perceived usefulness as "the degree to which a person believes that using a particular system would enhance his or her job performance". According to Venkatesh [28,29], perceived usefulness has a positive influence on the user's acceptance of various systems. An individual's behavioural intention to use systems is also positively affected by the perceived usefulness [16,31]. However, in the context of e-learning, perceived usefulness is redefined as "the extent to which a student believes that the e-learning system may help to improve his or her academic performance, by facilitating the whole learning process in general and the completion of learning-related tasks in particular" ([25], p. 302). Perceived usefulness is deeply embedded with factors including location/place, preferred time, feelings towards learning, and learning style when it comes to the e-learning systems [25]. Based on the findings of previous studies such as [25,31,33,41], we can postulate that higher the users' perceived usefulness in e-learning systems, their acceptance is more positive. Thus, we propose the following hypotheses:

**Hypothesis 5a (H5a).** *Perceived usefulness positively affects behavioural intention.*

**Hypothesis 5b (H5b).** *Perceived usefulness positively affects attitudes towards e-learning.*

2.2.2. External Factors
Facilitating Conditions

According to Salloum [42], "Facilitation Conditions" represent the physical resources that persuade an individual to complete certain tasks. The facilitation conditions are, however, subjective, and hence vary according to the perception of the individuals to work within certain systems [43]. A supportive external environment consist of availability of sufficient infrastructure and organisational resources [13,28]. Conversely, absence of favourable resources within an organisation impose bottlenecks for implementation of e-learning systems. The E-learning Acceptance Model introduced by Islam [44] validates technical support and online resources as key factors contributing to e-learning. They further identified availability of adequate number of computers, reliability of network, and access to online repositories as supportive conditions to promote e-learning. Individual usage of a system is largely determined by facilitation conditions [45]. Many scholars including Masádeh [46] and Tarhini [47] concluded the significant role of facilitation conditions on the acceptance of technology, thereby, the ease of use. Additionally, Venkatesh [29] and Venkatesh [45] also posit that facilitation conditions are pre-requisites of ease of use. Therefore, we formulate the following hypotheses:

**Hypothesis 6a (H6a).** *Facilitating condition has a positive relationship with actual use of e-learning.*

**Hypothesis 6b (H6b).** *Facilitating condition has a positive relationship with perceived ease of use.*

Subjective Norms

We adopt the definition constructed by Venkatesh [29] to define the Subjective Norm. Accordingly, Subjective Norm is "the person's perception that most people who are important to him/her think he/she should or should not perform the behaviour in question" ([29], p. 452). Agudo-Peregrina et al. [25] identified these important people as family and friends who exert a social pressure towards an individual. The originally developed TAM [11] emphasises the impact of Subjective Norm on behavioural intention through perceived usefulness, in the context of e-learning. Subjective Norm is referred to as the opinions of teachers, peers, and policies of educational institutions that convince the students towards the use of e-learning systems [9]. According to Abdullah [9], Subjective Norm is found to be a significant predictor of both the perceived ease of use and perceived usefulness of e-leaning among the students. Many other scholars also identified Subjective Norm as a strong predictor of perceived ease of use and perceived usefulness [16,22,48]. Based on the above discussion, we propose following hypotheses:

**Hypothesis 7a (H7a).** *Subjective norms positively affect behavioural intention.*

**Hypothesis 7b (H7b).** *Subjective norms positively affect attitudes towards e-learning.*

**Hypothesis 7c (H7c).** *Subjective norms positively affect perceived usefulness.*

Self-Efficacy

Self-efficacy as defined by Bandura [49] is an individual's evaluation of his or her own ability to perform a certain task. Self-efficacy judgement also determines how much effort individuals will put in and how long they will endure in the face of adversity. When faced with challenges, those who have significant doubts about their abilities either slacken their efforts or give up entirely, whereas those who have a strong sense of efficacy put up greater effort to overcome the obstacles [50]. Perceived self-efficacy predicts the degree of change in many forms of social behaviour. In the context of computer usage, computer self-efficacy is described as one's conviction in one's ability to execute a certain task using a computer [51]. Shen and Eder [51] showed that computer self-efficacy is a substantial predictor of perceived ease of use, followed by computer playfulness. Furthermore, they claimed that people who are more "playful" with technology find e-learning simpler to use, likely because they are willing to devote more time to learning new technology and find it entertaining. According to Abdullah [9], 33 out of 41 studies that they analysed confirmed the positive relationship between self-efficacy and perceived ease of use in the context of e-learning. Similarly, 10 out of 27 studies analysed reported a positive association between self-efficacy and perceived usefulness. This led us to formulate following hypotheses:

**Hypothesis 8a (H8a).** *Self-efficacy positively affects perceived usefulness.*

**Hypothesis 8b (H8b).** *Self-efficacy positively affects perceived ease of use.*

Relevance

Prior studies have identified relevance as a key factor in the acceptance of a system or a technology [25,41,52]. In the TAM3, Venkatesh [52] showed the correspondence between job relevance and the perceived usefulness of a system. The "job relevance" as defined by Venkatesh [52] (p. 191) is " ... an individual's perception regarding the degree to which the target system is applicable to his or her job". In the context of e-learning, we define relevance for learning as the learner's perception regarding the degree of importance of the learning system in performing the learning-related task. When a learner, i.e., a student, regards the use of the learning system as greatly important, the system could be seen as very useful. The relevance for learning had identified as the most important predictor of the perceived usefulness in [25]. In addition, in [41], major relevance (i.e., the relevance

of mobile learning for major studies) was found to have a significant relationship with perceived usefulness and attitudes towards mobile learning. Based on these prior findings, we propose the following hypotheses:

**Hypothesis 9a (H9a).** *Relevance positively affects perceived usefulness.*

**Hypothesis 9b (H9b).** *Relevance positively affects perceived ease of use.*

Computer Anxiety

Several studies have looked into computer anxiety as a crucial component in determining intention to use various technologies, such as e-mail and computer usage. Computer anxiety is described as the fear or apprehension that arises when a person is presented with the possibility of using a computer [53]. Those who are less anxious or worried are far more likely than those who are more anxious to interact with the information system. Because computer anxiety is mostly caused by a lack of past experience with an IS or a lack of self-confidence in efficiently managing or controlling it, anxiety over utilising a certain technology will have an impact on its adoption [22]. In higher education institutions, computer anxiety is a major factor in e-learning uptake, and this is due to the fact that people who are afraid of computers are more hesitant to use e-learning systems [9]. Piccoli [54] found that computer anxiety significantly affects the learner's satisfaction with an e-learning system. Lower levels of learning pleasure are associated with higher levels of computer anxiety. Computer anxiety has been found to influence both perceived ease of use and perceived usefulness [9]. Based on this prior evidence, we postulate a negative relationship between computer anxiety and both perceived usefulness and perceived ease of use. Thus, the following hypotheses are proposed:

**Hypothesis 10a (H10a).** *Computer anxiety negatively affects perceived usefulness.*

**Hypothesis 10b (H10b).** *Computer anxiety negatively affects perceived ease of use.*

Experience

Computer-related experience is defined as "the amount and type of computer skills a person acquires over time" ([55], p. 227). Individuals with greater computer-related experience, such as those who frequently use computers, the internet, and email, as well as have ability save and locate files, are more likely to have positive feelings about the ease of use and usefulness of an e-learning system [31]. Igbaria, Guimaraes, and Davis [56] suggested computer experience to be favourably related to system usage. User training and experience, which represent individual abilities and expertise, were found to be associated with user beliefs and usage. As a result, acceptability of computer technology was determined not only by the technology itself, but also by the skill level or expertise of the individual employing the technology [57]. Therefore, developing and acquiring technology-related knowledge, abilities, and attitudes can be a crucial prerequisite for e-learning system performance and adoption [57]. Accordingly, we propose the following hypotheses:

**Hypothesis 11a (H11a).** *Experience positive affects perceived usefulness.*

**Hypothesis 11b (H11b).** *Experience positively affects perceived ease of use.*

### 3. Material and Methods

*3.1. Survey Participants and Data Collection*

All graduate and undergraduate students in Sri Lankan State Universities were considered to be the population of this research. According to the statistical report issued by the UGC in Sri Lanka, the total enrolment of students in all 15 state universities by 2019 is approximately 424,620 [58]. However, due to the lack of a population framework, this study opted for a convenient sampling approach (i.e., a non-probabilistic approach)

following previous research in which this approach was widely adopted [13]. A Google-based online survey was administered to collected data from the students. The online survey was the most appropriate method of data collection due to prevailing COVID-19 outbreak in the country during the period of May to June 2020, when the data collection took place. The online questionnaire was presented to the students in several ways. The link to the Google-based questionnaire was posted in students' groups in social media such as Facebook, WhatsApp, and Viber. The questionnaire was also emailed to the students in different universities and requested them to circulate the questionnaire among their friends. The email addresses of the students were obtained from the deans, heads of the department and lecturers from each university upon formal request.

The sample size was determined on the basis of previous studies. According to the sample size recommendation of Cohen [59], the maximum number of arrows pointing to a construct in the structural model (Figure 1) is six and thus, the minimum sample required under the 99% confidence interval is 217. Similarly, previous studies have shown that a sample of at least 200 is sufficient for studies using Structural Equation Modelling (SEM) [60]. According to the formula $[n = N/(1 + N(e)^2)]$ suggested by Yamane [61] and retaining a 5% margin of error, the acceptable sample size for this study was approximately 400. However, 1039 students responded to the online survey, which is well above the minimum sample requirement (Table 1). There were no missing data or incomplete responses as all the questions in the online survey were compulsory and hence, incomplete questionnaires were not allowed to submit.

**Table 1.** Students' demographic characteristics (n = 1039).

| Variable | Particulars | Frequency | Percentage |
|---|---|---|---|
| Gender | Male | 349 | 33.6 |
| | Female | 690 | 66.4 |
| Year of study | First | 290 | 27.9 |
| | Second | 92 | 8.9 |
| | Third | 285 | 27.4 |
| | Fourth or Final | 372 | 35.8 |
| Area of Study | Agriculture | 112 | 10.8 |
| | Applied Sciences | 132 | 12.7 |
| | Arts | 128 | 12.3 |
| | Engineering | 33 | 2.2 |
| | Management | 506 | 48.7 |
| | Medicine | 26 | 2.5 |
| | Technology | 112 | 10.8 |

### 3.2. Survey Instrument

The questionnaire was designed with close-ended questions in three sections. The first section intended to obtain demographic information from the respondents including gender, year of study, and area of study. The second section contained 48 items used to measure latent constructs, except for the actual use of e-learning (Figure 1). The last section contained four items intended to capture self-reported actual use of e-learning (i.e., number of minutes or hours per day and number of days per week).

The measurement scales for the latent construct were drawn from previous studies, and minor modifications were carried out where necessary. As the theoretical model developed in this study is an extension of GETAMEL [9], primary constructs of the model, namely, Self-Efficacy (SE), Computer Anxiety (CA), Experience (EX), Subjective Norms (SN), Facilitating Conditions (FC), Perceived Usefulness (PU), Perceive Ease of Use (PE), Attitudes (AT), and Behavioural Intention (BI), were adopted from the related studies [9,13,16,17,22,31,62]. Accordingly, the measurement scales of SE, CA, and BI contained three items each and EX, PE, and SN had four items each. PU was measured with five items, and FC was measured using ten items. The latent construct RE was measured using five items following Park et al. [22]. The moderating variable of the model, Precipitating Events (CV), was

measured using a newly developed scale consisting of three items. However, this scale was somewhat identical to the scale developed by Moghavvemi et al. [35] for the measurement of precipitating events.

Since we incorporated a few newly developed scales into the survey instrument, a pilot study was conducted to ensure the reliability and validity of the questionnaire. Before the pilot test, however, two professors were consulted, and several modifications were carried out in the wording of the questionnaire. Subsequently, the questionnaire was sent out to the students of two universities: the Rajarata University of Sri Lanka and the Uva Wellassa University of Sri Lanka. In the pilot test, we received a total of 105 completed questionnaires from both the universities. Following a reliability test based on Cronbach's Alpha, the internal consistency reliability for all the measurement items was confirmed, except for Actual Usage of e-learning (AU). Further modifications were carried out in the measurement scale of AU prior to the final survey (see the Supplementary Materials).

### 3.3. Data Analysis

This research aims to examine the factors that determine the use of e-learning and to investigate the effects of COVID-19 pandemic on the use of e-learning among university students. This study also attempts to assess the degree to which these factors influence the use of e-learning. Previous studies have shown that the use of the Partial Least Square Structural Equating Model (PLS-SEM) is better suited for confirming key factors and predicting and explaining target constructions in an empirical model [63]. Therefore, we used a PLS-SEM approach using Smart PLS 3 software in our data analysis and model evaluation. More recently, PLS-SEM has widely been used in various studies in different disciplines such as Total Quality Management, IT, Environmental Accounting, Finance, and particularly in Marketing, relative to covariance-based SEM (CB-SEM) (see, for example, [64–67]). This is because PLS-SEM is capable of handling complex models, and because of its high statistical power and flexibility in parametric assumption and sample size [68,69].

We used the PLS-SEM approach in the following manner. First, we established the relevant measurement model criteria using the procedure recommended by [68]. Because the indicators are reflectively defined, we assessed the internal consistency reliability, convergent validity, and discriminant validity of the measurement model using standard criteria such as Cronbach's Alpha, composite reliability (CR), average variance extracted (AVE), Fornell–Larcker Criterion, and Heterotrait-Monotrait Ratio (HTMT) [68,70]. Second, we estimated a few standard parameters, such as the variance inflation factor (VIF), R-square ($R^2$), and cross-validated redundancy ($Q^2$), in order to evaluate the structural model [63,64]. Following [70], we also computed effect size $f^2$, which determines the effect of removing a specific exogenous variable on the $R^2$ of each endogenous variable. Finally, we evaluated the hypotheses using path coefficients, t-statistics, and *p*-values. We also used bootstrapping with 5000 sub-samples to calculate t-statistics and *p*-values [68,69].

## 4. Results

Tables 2–5 report the results relating to the assessment of the measurement model. The descriptive statistics for each indicator and their outer loadings are presented in Table 2. The mean value of most of the indicators is centred around 5, except for the indicators relating to CA. Concerning the external factors (i.e., SE, SN, RE, FC, EX, CV), the mean value of 5 can be interpreted as that the majority of students have above average agreement with each statement (indicator) under each construct. This indicates that majority of the respondents (i.e., university students) believe that they are fairly capable of handling the e-learning related task themselves. Moreover, the majority of students believe that e-learning is important for their studies and they are confident using e-learning for their studies. Moreover, higher mean values for indicators relating to FC indicate that the students have fairly sufficient facilitating conditions for the e-leaning. However, affordability of internet charges and availability of equipment such as laptops, tabs, and smartphones seem to be unfavourable to students e-learning in Sri Lanka.

**Table 2.** Summary statistics and indicators outer loading.

| Latent Constructs | Indicators | Loadings | Mean | Median | Std. Dev. | Excess Kurtosis | Skewness |
|---|---|---|---|---|---|---|---|
| Self-Efficacy (SE) | SE1 | 0.891 *** | 5.25 | 5 | 1.528 | −0.137 | −0.67 |
| | SE2 | 0.882 *** | 5.113 | 5 | 1.607 | −0.225 | −0.674 |
| | SE3 | 0.868 *** | 4.804 | 5 | 1.565 | −0.583 | −0.331 |
| Relevance (RE) | RE1 | 0.881 *** | 5.594 | 6 | 1.495 | 0.486 | −1.033 |
| | RE2 | 0.892 *** | 5.709 | 6 | 1.448 | 0.594 | −1.11 |
| | RE3 | 0.893 *** | 5.369 | 6 | 1.409 | 0.163 | −0.744 |
| | RE4 | 0.812 *** | 5.406 | 6 | 1.495 | 0.122 | −0.839 |
| | RE5 | 0.843 *** | 5.696 | 6 | 1.373 | 0.88 | −1.084 |
| Computer Anxiety (CA) | CA1 | 0.961 *** | 2.869 | 2 | 1.653 | −0.607 | 0.58 |
| | CA2 | 0.968 *** | 2.818 | 2 | 1.69 | −0.571 | 0.667 |
| | CA3 | 0.835 *** | 2.914 | 3 | 1.676 | −0.757 | 0.515 |
| Experience (EX) | EX1 | 0.844 *** | 5.473 | 6 | 1.549 | 0.267 | −0.934 |
| | EX2 | 0.865 *** | 5.332 | 6 | 1.634 | −0.211 | −0.819 |
| | EX3 | 0.904 *** | 5.603 | 6 | 1.496 | 0.588 | −1.085 |
| | EX3 | 0.906 *** | 5.441 | 6 | 1.478 | 0.368 | −0.938 |
| Subjective Norms (SN) | SN1 | 0.854 *** | 5.312 | 5 | 1.365 | −0.053 | −0.607 |
| | SN2 | 0.883 *** | 5.088 | 5 | 1.506 | −0.129 | −0.581 |
| | SN3 | 0.904 *** | 5.145 | 5 | 1.427 | −0.205 | −0.527 |
| | SN4 | 0.910 *** | 5.269 | 5 | 1.376 | 0.106 | −0.637 |
| Perceived Usefulness (PU) | PU1 | 0.886 *** | 5.267 | 5 | 1.424 | −0.001 | −0.645 |
| | PU2 | 0.934 *** | 5.435 | 6 | 1.344 | 0.282 | −0.76 |
| | PU3 | 0.931 *** | 5.361 | 6 | 1.356 | 0.192 | −0.695 |
| | PU4 | 0.926 *** | 5.334 | 5 | 1.383 | 0.373 | −0.764 |
| | PU5 | 0.906 *** | 5.655 | 6 | 1.338 | 0.795 | −1.004 |
| Perceived Ease of Use (PE) | PE1 | 0.880 *** | 5.457 | 6 | 1.358 | 0.451 | −0.851 |
| | PE2 | 0.915 *** | 5.301 | 5 | 1.323 | 0.051 | −0.583 |
| | PE3 | 0.910 *** | 5.199 | 5 | 1.357 | −0.066 | −0.531 |
| | PE4 | 0.909 *** | 5.262 | 5 | 1.414 | −0.077 | −0.619 |
| Facilitating Condition (FC) | FC1 | 0.694 *** | 4.461 | 5 | 1.808 | −0.916 | −0.279 |
| | FC2 | 0.348 * | 4.038 | 4 | 1.768 | −0.891 | −0.05 |
| | FC3 | 0.379 * | 5.158 | 3 | 1.75 | −0.413 | −0.735 |
| | FC4 | 0.855 *** | 5.417 | 6 | 1.575 | 0.118 | −0.921 |
| | FC5 | 0.864 *** | 5.269 | 6 | 1.627 | −0.282 | −0.771 |
| | FC6 | 0.812 *** | 5.149 | 5 | 1.486 | −0.155 | −0.62 |
| | FC7 | 0.820 *** | 5.189 | 5 | 1.502 | −0.046 | −0.691 |
| | FC8 | 0.836 *** | 5.101 | 5 | 1.544 | −0.164 | −0.685 |
| | FC9 | 0.832 *** | 4.977 | 5 | 1.502 | −0.225 | −0.53 |
| | FC10 | 0.786 *** | 4.972 | 5 | 1.74 | −0.386 | −0.686 |
| Attitudes Towards e-learning (AT) | AT1 | 0.948 *** | 5.48 | 6 | 1.413 | 0.424 | −0.898 |
| | AT2 | 0.955 *** | 5.473 | 6 | 1.384 | 0.509 | −0.894 |
| | AT3 | 0.959 *** | 5.555 | 6 | 1.349 | 0.536 | −0.899 |
| | AT4 | 0.936 *** | 5.441 | 6 | 1.355 | 0.093 | −0.723 |
| Behavioural Intention to Use e-learning (BI) | BI1 | 0.937 *** | 5.279 | 5 | 1.383 | 0.089 | −0.697 |
| | BI2 | 0.942 *** | 5.291 | 5 | 1.35 | 0.056 | −0.637 |
| | BI3 | 0.904 *** | 5.56 | 6 | 1.39 | 0.555 | −0.966 |
| Precipitating Events (CV) | CV1 | 0.923 *** | 5.357 | 6 | 1.653 | 0.192 | −0.974 |
| | CV2 | 0.943 *** | 5.436 | 6 | 1.559 | 0.49 | −1.025 |
| | CV3 | 0.918 *** | 5.262 | 6 | 1.643 | −0.052 | −0.841 |
| Actual use of e-learning (AU) | AU1 | 0.395 ** | 3.727 | 4 | 1.337 | 1.158 | −0.56 |
| | AU2 | 0.840 *** | 5.216 | 5 | 1.547 | 0.986 | −0.984 |
| | AU3 | 0.692 *** | 4.556 | 5 | 1.823 | −1.169 | −0.157 |
| | AU4 | 0.901 *** | 4.783 | 5 | 1.939 | −1.12 | −0.382 |

Note: Significant at *** $p > 0.001$, ** $p > 0.01$, * $p > 0.05$.

**Table 3.** Internal consistency reliability and convergent validity.

| Latent Constructs | Cronbach's Alpha | rho_A | CR | AVE |
|:---:|:---:|:---:|:---:|:---:|
| AT | 0.964 | 0.964 | 0.974 | 0.902 |
| AU | 0.744 | 0.775 | 0.855 | 0.665 |
| BI | 0.919 | 0.919 | 0.949 | 0.861 |
| CA | 0.912 | 0.928 | 0.945 | 0.853 |
| CV | 0.919 | 0.921 | 0.949 | 0.861 |
| EX | 0.903 | 0.907 | 0.932 | 0.775 |
| FC | 0.927 | 0.930 | 0.940 | 0.663 |
| PE | 0.925 | 0.925 | 0.947 | 0.816 |
| PU | 0.952 | 0.953 | 0.963 | 0.841 |
| RE | 0.915 | 0.918 | 0.937 | 0.748 |
| SN | 0.910 | 0.914 | 0.937 | 0.788 |
| SE | 0.855 | 0.858 | 0.912 | 0.775 |

Notes: CR, Composite Reliability; AVE, Average Variance Ext.

*4.1. The Assessment of the Measurement Model*

The factor loadings of all the indicators are well above the recommended level of 0.7 [63,68], excepts for the two indicators related FC, as shown in the Table 2. These two indicators (FC2 and FC3) were excluded from the structural model evaluation. Internal consistency reliability of the measurement model was assessed referring to Cronbach's Alpha and CR, the values of which exceed the recommended criterion level of 0.7 (see Table 3). The satisfactory values of Cronbach's Alpha and CR indicate that the respective constructs are well measured by the chosen indicators. The AVE values above 0.5 for all latent constructs indicate a satisfactory level of convergent validity. Concerning the discriminant validity, we used three metrics: Fornell–Larker Criterion, cross-loadings, and Heterotrait-Monotrait Ratio (HTMT) [68,69]. According to the Fornell–Larker Criterion, the square root of the AVE of each latent construct should be higher than its correlation with other construct in the model. On the other hand, an HTMT ratio of below 0.85 is considered to be the accepted level of convergent validity [68,70]. Accordingly, as shown in Tables 4 and 5, both the Fornell–Larker Criterion and HTMT indicate adequate levels of discriminant validity. Untabulated cross loadings also show appropriate level of discriminate validity. We also assessed the multi-collinearity of exogenous latent constructs with respect to their endogenous latent construct in the structural model and the results of which is presented in Table 6. The VIF values below 5 are indicative of the absence of multi-collinearity issue in the empirical model [66].

**Table 4.** Fornell–Larcker Criterion for discriminant validity.

| Latent Constructs | AT | AU | BI | CA | CV | EX | FC | PE | PU | RE | SN | SE | DV Met |
|---|---|---|---|---|---|---|---|---|---|---|---|---|---|
| AT | 0.950 | | | | | | | | | | | | Yes |
| AU | 0.352 | 0.816 | | | | | | | | | | | Yes |
| BI | 0.892 | 0.359 | 0.928 | | | | | | | | | | Yes |
| CA | −0.299 | −0.164 | −0.304 | 0.924 | | | | | | | | | Yes |
| CV | 0.483 | 0.392 | 0.490 | −0.161 | 0.928 | | | | | | | | Yes |
| EX | 0.704 | 0.345 | 0.709 | −0.357 | 0.422 | 0.880 | | | | | | | Yes |
| FC | 0.747 | 0.323 | 0.734 | −0.281 | 0.420 | 0.678 | 0.814 | | | | | | Yes |
| PE | 0.826 | 0.328 | 0.802 | −0.324 | 0.465 | 0.753 | 0.757 | 0.903 | | | | | Yes |
| PU | 0.845 | 0.346 | 0.804 | −0.282 | 0.465 | 0.720 | 0.715 | 0.870 | 0.917 | | | | Yes |
| RE | 0.795 | 0.316 | 0.757 | −0.235 | 0.436 | 0.678 | 0.646 | 0.757 | 0.806 | 0.865 | | | Yes |
| SN | 0.728 | 0.289 | 0.696 | −0.225 | 0.394 | 0.594 | 0.663 | 0.747 | 0.720 | 0.655 | 0.888 | | Yes |
| SE | 0.626 | 0.276 | 0.639 | −0.266 | 0.408 | 0.637 | 0.621 | 0.680 | 0.651 | 0.653 | 0.506 | 0.880 | Yes |

Notes: Values on the diagonal are square root of AVE. The rest of values are correlation of each latent variable with other latent variables. DV, Discriminant Validity.

**Table 5.** Heterotrait-Monotrait Ratio for discriminant validity.

| Latent Constructs | AT | AU | BI | CA | CV | EX | FC | PE | PU | RE | SN | SE | DV Met |
|---|---|---|---|---|---|---|---|---|---|---|---|---|---|
| AU | 0.415 | | | | | | | | | | | | |
| BI | 0.448 | 0.429 | | | | | | | | | | | Yes |
| CA | 0.317 | 0.190 | 0.331 | | | | | | | | | | Yes |
| CV | 0.512 | 0.470 | 0.533 | 0.174 | | | | | | | | | Yes |
| EX | 0.753 | 0.410 | 0.777 | 0.394 | 0.462 | | | | | | | | Yes |
| FC | 0.788 | 0.387 | 0.793 | 0.302 | 0.453 | 0.740 | | | | | | | Yes |
| PE | 0.775 | 0.388 | 0.769 | 0.351 | 0.504 | 0.822 | 0.815 | | | | | | Yes |
| PU | 0.782 | 0.406 | 0.760 | 0.302 | 0.496 | 0.774 | 0.760 | 0.826 | | | | | Yes |
| RE | 0.845 | 0.378 | 0.824 | 0.256 | 0.475 | 0.743 | 0.697 | 0.821 | 0.763 | | | | Yes |
| SN | 0.775 | 0.351 | 0.759 | 0.245 | 0.428 | 0.652 | 0.719 | 0.813 | 0.770 | 0.714 | | | Yes |
| SE | 0.687 | 0.336 | 0.719 | 0.301 | 0.459 | 0.722 | 0.695 | 0.763 | 0.720 | 0.734 | 0.571 | | Yes |

Note: DV, Discriminant Valid.

**Table 6.** The assessment of multi-collinearity.

| Latent Constructs | AT | AU | BI | PE | PU |
|---|---|---|---|---|---|
| AT | | | 4.195 | | |
| AU | | | | | |
| BI | | 2.373 | | | |
| CA | | | | 1.154 | 1.163 |
| CV | | 1.330 | | | |
| EX | | | | 2.514 | 2.659 |
| FC | | 2.191 | | 2.227 | |
| PE | 4.093 | | 3.030 | | 4.216 |
| PU | 4.300 | | 3.234 | | |
| RE | | | | 2.329 | 2.780 |
| SN | 2.371 | | 2.502 | | 2.378 |
| SE | | | | 2.102 | 2.130 |

Note: This table reports inner Variance Inflation Factors (VIF) of the exogenous latent constructs with respect to their endogenous latent constructs for the assessment of multicollinearity. All the VIF values reported in the table above are indicative of the absence of multicollinearity, since none of values exceed the maximum threshold of 5.

*4.2. Hypotheses Testing*

The evaluation of the structural model involves three parameters being estimated and validated: goodness of fit (GoF), path coefficients, and coefficient of determination ($R^2$). The overall quality of the empirical model, i.e., GOF, was determined by using the equation employed by Chao [17]: GoF = $\sqrt{\overline{AVE} \times \overline{R}^2}$. In this study, the value derived from this equation for GoF was 0.72, which was well above the benchmark value of 0.36 [71], indicating that the research model has well fitted to the data. However, overall model fit is trivial since we were examining the hypothesised relationship represented by each path in the model by using the PLS-SEM approach rather than testing a single theory. Finally, we assessed the hypothesised relationships referring to path coefficient and corresponding *p*-value and t statistics.

Table 7 reports the results of the structural model evaluation. Our empirical model developed in this study aims to test 21 hypotheses, including sub-hypotheses. The first five hypotheses (i.e., H1–H5) are related to core predictors of e-learning use behaviour and moderating effect of precipitating events (i.e., H2). The rest of the hypotheses theorise the relation between external factors and the core predictors of the empirical model.

Consistent with the behavioural theories and technology acceptance models, we predicted a positive relationship between BI and AU. Our results, similar to those of many previous studies [9,13,16,17,25], indicate a significant and positive relationship between BI and AU (β = 0.158, *p* < 0.01), which supported our prediction in H1. However, the COVID-19 pandemic and related events, precipitating events, did not appear to moderate the relationship between BI and AU, contrary to our prediction in H2. Precipitating events not only had a non-significant moderating effect, but they also had a negative moderation as opposed to our prediction (β = −0.028, *p* > 0.05). Nonetheless, the direct impact of the COVID-19 pandemic-related events on e-learning use was highly significant (β = 0.278, *p* < 0.001). This suggests that students who believed COVID-19 affected their use of e-learning has actually used e-learning to a larger degree.

The strongest bivariate relationship (i.e., the largest path coefficient) in the model was found in between AT and BI (β = 0.682, *p* < 0.001), which supported the H3. Concerning the other predictors of BI, only PE had a significantly positive effect on BI (β = 0.149, *p* < 0.001), thereby supporting H4a. Non-significant relationship between PU and BI rejected H5a, and similarly H7a was rejected as the path coefficient between SN and BI was not significant. Notably, all predictors of AT (i.e., PU, PE, and SN) were found significant supporting H4b, H5b, and H7b (β = 0.284, β = 0.472, β = 0.177, *p* < 0.001). Among the predictors of AT, PU was shown to have the largest effect on the AT ($f^2$ = 0.217). On the other hand, our results indicate PE had a strong positive relationship and the largest effect on PU, which provide

enough evidence to accept H4c. Meanwhile, the relationship between SN and PU was also found significant, supporting H7c.

**Table 7.** Hypothesis testing.

| Hypotheses | Path | Path Coefficients | S.E. | T-Stat. | *p*-Values | $f^2$ | Decision |
|---|---|---|---|---|---|---|---|
| H1 | BI → AU | 0.158 | 0.047 | 3.061 | 0.001 ** | 0.031 | Accept |
| H2 | BI * CV → AU | −0.028 | 0.026 | 1.079 | 0.281 | 0.001 | Reject |
| | CV → AU | 0.278 | 0.044 | 6.302 | 0.000 *** | 0.072 | |
| H3 | AT → BI | 0.682 | 0.033 | 20.792 | 0.000 *** | 0.586 | Accept |
| H4a | PE → BI | 0.149 | 0.039 | 3.804 | 0.000 *** | 0.023 | Accept |
| H4b | PE → AT | 0.284 | 0.041 | 6.843 | 0.000 *** | 0.072 | Accept |
| H4c | PE → PU | 0.511 | 0.039 | 12.929 | 0.000 *** | 0.331 | Accept |
| H5a | PU → BI | 0.073 | 0.037 | 1.947 | 0.052 | 0.005 | Reject |
| H5b | PU → AT | 0.472 | 0.044 | 10.840 | 0.000 *** | 0.217 | Accept |
| H6a | FC → AU | 0.088 | 0.043 | 2.016 | 0.054 | 0.004 | Reject |
| H6b | FC → PE | 0.305 | 0.032 | 9.552 | 0.000 *** | 0.163 | Accept |
| H7a | SN → BI | 0.036 | 0.027 | 1.316 | 0.188 | 0.003 | Reject |
| H7b | SN → AT | 0.177 | 0.030 | 5.878 | 0.000 *** | 0.055 | Accept |
| H7c | SN → PU | 0.092 | 0.025 | 3.751 | 0.000 *** | 0.019 | Accept |
| H8a | SE → PU | 0.017 | 0.025 | 0.680 | 0.497 | 0.001 | Reject |
| H8b | SE → PE | 0.126 | 0.032 | 4.013 | 0.000 *** | 0.030 | Accept |
| H9a | RE → PU | 0.305 | 0.032 | 9.686 | 0.000 *** | 0.180 | Accept |
| H9b | RE → PE | 0.301 | 0.029 | 10.484 | 0.000 *** | 0.151 | Accept |
| H10a | CA → PU | 0.003 | 0.015 | 0.190 | 0.849 | 0.001 | Reject |
| H10b | CA → PE | −0.046 | 0.017 | 2.673 | 0.008 *** | 0.007 | Accept |
| H11a | EX → PU | 0.064 | 0.028 | 2.331 | 0.020 ** | 0.008 | Accept |
| H11b | EX → PE | 0.245 | 0.031 | 7.962 | 0.000 *** | 0.093 | Accept |

Notes: The results of hypothesis testing reported in this table is based on bootstrapping method using 5000 subsamples. S.E., Standard Error; T-Stat., T-Statistics. $f^2$, $f$ square measures the effect size of removal of each endogenous variable on the $R^2$ of respective exogenous variable. As a rule of thumb, $f^2$ values of 0.02, 0.15, and 0.35 are indicative of small, medium, and large effects [63]. Significant at *** $p > 0.001$, ** $p > 0.01$, * $p > 0.05$.

With respect to external factors, we obtained mixed findings. As shown in Table 7 and depicted in Figure 2, FC had a significant effect on PE and yet a non-significant effect on AU. Therefore, we could only accept H6b and reject H6a. RE and EX were significantly and positively associated with both the PU and PE; this is consistent with our prediction in H9a, H9b, H11a, and H11b (β = 0.305, 0.301, 0.064, 0.245, $p < 0.001$, $p < 0.05$). Moreover, we found PU and PE were substantially predicted by the RE; that is, an increase in one standard deviation in RE increases 30.5% in PU and 30.1% in PE. Furthermore, SE had a positive and significant relationship with PE and non-significant relationship with PU, which supported H8b but rejected H8a. Concerning CA, our findings indicated a negative and significant relationship between CA and PE (β = −0.046, $p < 0.01$), and a non-significant relation with PU.

Except for AU, the empirical model adequately explains all of the endogenous variables in terms of coefficient of determination ($R^2$): 81.1% of BI, 76.2% of AT, 81.3% of PU, and 74.4% of PE. However, the overall model explanatory power is 19.5%, indicating that the components included in the model do not explain a significant portion of the variance in AU, which is the ultimate endogenous variable. Nonetheless, our empirical model surpasses Falk and Miller's [72] minimum criteria for coefficient of determination ($R^2$); according to them, the coefficient of determination ($R^2$) should be greater than 0.10. With regards to predictive accuracy and relevance of the path coefficients, the $Q^2$ values of AU, BI, AT, PE, and PU (0.126, 0.693, 0.683, 0.602, and 0.679, respectively) indicate a satisfactory level.

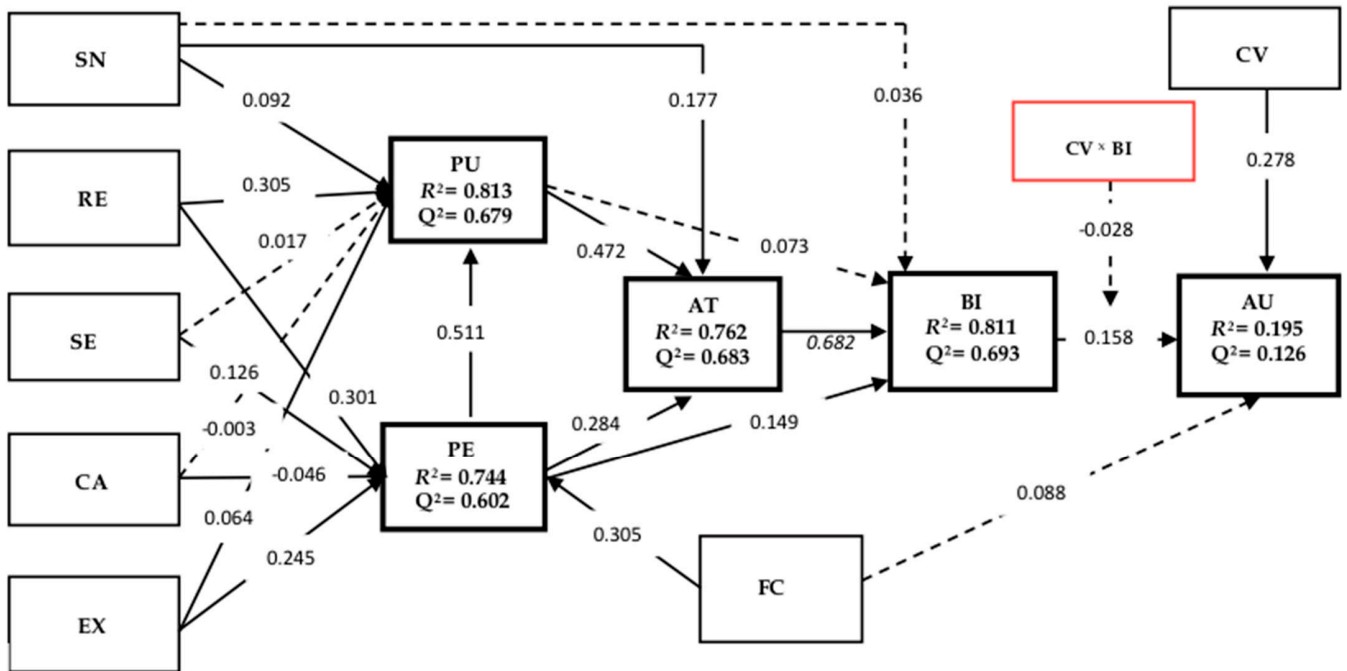

**Figure 2.** Results of the evaluation of structural model using PLS-SEM. Note: This figure depicts the results of structural model evaluation. $R^2$, or the coefficient of determination, is the amount of variation in each endogenous variable explained by its exogenous factors. $Q^2$ measures the predictive accuracy and relevance of the path coefficients. All the strait arrows indicate accepted hypotheses and dashed arrows indicate rejected hypotheses.

## 5. Discussion and Implications

This study sought to answer two research questions. First, we examined what factors impel or impede Sri Lankan undergraduates' AU of e-learning. Second, we investigated how the COVID-19 pandemic and its consequences affect the use behaviour of e-learning among undergraduates in Sri Lanka. The empirical model in this study was based primarily on the TAM and included a number of other external variables such as FC, EX, CA, SE, and SN. Drawing on the entrepreneurial potential model and conceptualising through precipitating events, the moderating effect of the COVID-19 pandemic and associated events on the relationship between use intention and use behaviour was also examined in this study. The empirical model of this study is unique in that it incorporates the effect of the COVID-19 pandemic and related events through a moderating variable. The model is also considerably more comprehensive than other traditional models (i.e., TAM, UTAUT) since it incorporates the majority of the variables that have been shown to be significant in evaluating the intention and actual use of e-learning. We tested our empirical model on a relatively large sample of university students (n = 1039), and to the best of our knowledge, this is the largest e-learning survey ever conducted in Sri Lanka. The empirical findings of this study are mostly consistent with previous studies, in particular, the relationship between core predictors of e-learning. Nevertheless, our findings do not provide sufficient evidence to support our prediction regarding the moderating impact of the COVID-19 pandemic and associated events on the intention–behaviour relationship. The following section provides further interpretation of our empirical findings.

The intention–behaviour relationship has often been tested in many studies in the IS literature [9,18,19,32,34] and behavioural literature in general; this relationship has always been found to be significant, except for a few studies (for insignificant relationship, see [25]). Although the behavioural intention is presumed to be the leading cause of the actual behaviour, as per most of the behavioural theories (i.e., TRA, TPB, TAM, UTAUT), previous empirical findings, particularly in the IS literature, suggest otherwise. In the context of e-learning, BI or use intention has repeatedly been reported as less important in

explaining the variation in the actual behaviour; that is, BI can explain less than half of the variance of the actual behaviour [25]. This is also evident in the strength of the relationship between BI and AU of e-learning. Many studies have reported a small or moderate level of path coefficients in between BI and AU of e-learning (see, for example, [13,25,48]). Our study is no exception; we found a relatively smaller association between BI and AU of e-learning with respect to the undergraduates in the Sri Lankan universities. Our finding with regards to the link between BI and AU of e-learning is somewhat consistent with a similar study conducted in Sri Lanka by [13], where they also reported a small path coefficient between BI and AU of e-learning. However, the lower association between BI and AU of e-learning in our study could be attributed to the sample profile, which comprises a fairly large number of students from different study disciplines. According to the responses, we observed that students in different majors had a substantial heterogeneity in the use of e-learning regardless of their use intention.

Contrary to our prediction, the moderating effect of the COVID-19 pandemic and related events, conceptualised as precipitating events, on the intention–behaviour relationship found to be insignificant. Although the majority of students felt that the pandemic and associated events had largely influenced their decision to use or increase their usage of e-learning, the interaction effect of AU and CV appears to have a little or no impact on AU of e-learning. This indicates that the association between BI and AU of e-learning is independent of students' perception of how the present health crisis and its consequences influenced their usage of e-learning. These findings, however, might have been different if data had been collected during the second and third waves of the COVID-19 pandemic, which occurred in late 2020 and early 2021 in Sri Lanka. We collected the data a few months after the country's first COVID-19 outbreak, and most individuals, including university students, were unsure about their future actions during that time. Although the students believed that the COVID-19 pandemic affected their decision to use or expand their use of e-learning, their actual usage may have been hindered by the ambivalence caused by the pandemic. Still, the COVID-19 pandemic and related events had a positive and significant impact on the AU of e-learning; this is similar to the findings of [73], where they observed drastic increase in students participation in Moodle courses following the COVID-19 outbreak.

Surprisingly, BI to use e-learning was well explained by its predictors: attitudes towards e-learning, perceived usefulness, perceived ease of use, and subjective norms. In other words, the TAM framework provides a much better explanation for university students' BI to use e-learning in the Sri Lankan context. In contrast to previous studies, the exogenous variables (i.e., AT, SN, PU, and PE) explained the majority of the variance in BI ($R^2 = 0.811$). The reported $R^2$ for BI is much greater than that of previous studies related to e-learning (see, for example, [13,17,25]). Out of the four predictors of BI, the highest path coefficient, thereby the strongest relationship, was found between AT and BI, and on the other hand, the weakest relationship was found between SN and BI. While this suggests the attitudes of the students towards e-learning is the best predictor of BI to use e-learning, this is inconsistent with the findings of [16,41], where they found SE and SN to be the best predictors of BI. This has important implications for the Sri Lankan higher education sector. When promoting e-learning, administrators of the universities or other educational institution should be aware of the present condition of students' attitudes towards e-learning and strategies for fostering favourable attitudes toward e-learning in order to successfully adopt e-learning systems.

Our empirical model could be able to explain a substantial portion of the variance in AT, which is evident in the coefficient of the determination ($R^2 = 0.762$). All the antecedents of AT, i.e., PU, PE, and SN, were found to be significant. However, AT seems to have a stronger correlation with PU, followed by PE. This signifies that students who viewed e-learning to be beneficial and easy to use are more likely to have favourable attitudes towards it. Our findings support the core assumptions of behavioural theories such as TRA and TPB, which hold that attitude formation is the consequence of a person's evaluation or



appraisal of the outcomes of the behaviour in question [26]. When students appraise the consequences of e-learning favourably, that is, when e-learning provides them with benefits such as enhanced study performance and learning efficiency and less effort in finding suitable and relevant study materials, and they perceive these outcomes as desirable, they develop positive attitudes toward e-learning. This is critical in designing and implementing e-learning systems in higher education institutes such as in universities because educators should be able to persuade students that using e-learning systems will result in desirable outcomes, such as higher grades. Our findings also imply that people who are significant to students play a vital role in shaping their attitudes toward e-learning. According to the students' responses, teachers are the most significant referent group whose views are most likely to influence their attitudes on e-learning. Indeed, amid this global health crisis, teachers have a significant amount of responsibility for making e-learning as effective as traditional learning [2]. Learning and assessment methods should be revisited so that everyone can participate and benefit. Our findings with regards to AT and its predictors are well aligned with the findings of previous studies [10,22].

Turning to external variables, the results indicate that SE, RE, EX, CA, and FC are significantly associated with PE and/or PU. Relevance of e-learning to students' major studies found a positive and significant relationship with both the PE and PU. Students' experience using computers also had a positive and significant relationship with PE and PU. These findings echo similar findings in [16,22]. Our sample comprises a majority of students from management and technology faculties, and almost all the courses in these faculties can be taught and learn online. This may be the reason why we observe a strong positive association of RE and EX with PE and PU, as the students in these faculties find e-learning to be more relevant and enjoyable.

Overall, the majority of hypothesised relationships in this study were accepted, demonstrating the reliability and robustness of the empirical model in evaluating e-learning usage. Of the relationship conjectured in this study, the moderating effect of the COVID-19 pandemic on the intention–behaviour relationship was found to be non-significant. However, when compared to the prior models, our empirical model appears to be superior in predicting BI to use of e-learning. This study's findings also underline the significance of students' AT, PU, and PE in predicting the BI and AU of e-learning. Similarly, external variables such as EX, RE, and FC appear to play a significant role in e-learning usage.

## 6. Concluding Remarks

The COVID-19 pandemic has led to an unprecedented transformation in the education sector, with teaching and learning shifting entirely online. While this has presented many challenges to both educators and students, considerable efforts have been undertaken to address these difficulties. The present study attempted to explore the effects of the COVID-19 pandemic on e-learning usage among undergraduate students in Sri Lankan universities. This study also assessed the key factors of e-learning usage and provides implications for the higher education sector. We collected the data from a sample of 1039 students from several state universities in Sri Lanka and conducted our analysis using a comprehensive empirical model based on the TAM framework. Despite the fact that the COVID-19 pandemic and related events had no significant moderating effect on the behaviour–intention relationship, the actual use of e-learning appears to have been strongly impacted by the COVID-19 pandemic. Moreover, in line with the mainstream literature, our findings indicate that behavioural intention is a significant predictor of e-learning usage. On the other hand, attitudes toward e-learning and perceived ease of use were shown to be important predictors of behavioural intention. Our findings also imply that subjective norms, perceived usefulness, and perceived ease of use can effectively predict attitudes towards e-learning. Finally, relevance for learning and experience using computers were found to be closely associated with perceived usefulness and perceived ease of use.

The findings of this study highlighted that although behavioural intention was crucial for e-learning usage, the actual usage of e-learning seemed to be determined by many

factors unaccounted for by the extant behavioural and acceptance models. These factors may be varied over the time, place, and context to context, which poses a real challenge in identifying them. This opens up further avenues for academia to explore and extend the current behavioural and acceptance models. Our findings also provide important implications for educators, administrators, and other officials in the higher education sector. Since attitude stands out as an important factor, students' attitudinal change is paramount in the effectiveness of e-learning usage. This must be taken into account in designing and implementing e-learning systems.

Our study shed light on the effects of the COVID-19 pandemic on the teaching and learning process. The findings of this study can help both educators and students to foresee the potential impact of the pandemic and realign the pedagogical and evaluation methods. The educational policymakers are also benefitted through the study as the study uncovered critical factors impacting the use of e-learning. Therefore, educators and policymakers can promote online learning in an effective way by focussing on these critical factors. The external factors such as relevance, experience, facilitating conditions, and subjective norms were found to be crucial since these factors significantly affect the perceived ease of use and perceived usefulness of e-learning. Innovative teaching and learning techniques based on technology should be implemented in order to increase students' perceptions of usefulness and ease of use. The involvement of university teachers is essential in this regard, since the findings indicate that students see teachers as important parties who shape their attitudes toward e-learning.

We acknowledge the following limitations in our study. First, although our empirical model surpassed the required statistical parameters satisfactorily, the model can explain only a small fraction of variance in the actual usage of e-learning. This could be due to either the ineffectiveness of measurement scales of actual usage of e-learning or the variations in the actual usage of e-learning by different groups of students. Despite having increased use intention, students in various fields of study may not use e-learning to an equal extent for their major studies, and hence, the actual usage is varied between the groups of students. A multi-group analysis, in which the empirical model is estimated separately for each group, can detect heterogeneity in the use of e-learning among different student groups (i.e., Art, Commerce, IT, Engineering, and Medicine), as well as its relationship with the other variables in the model (i.e., group-specific path coefficients). However, we leave this for future studies. Second, our study relied on self-reported data to measure the actual usage of e-learning. Since the self-reported data may be noisy due to a variety of reasons, including the arbitrariness of students' responses, the actual usage of e-learning may not be reflected in the data. Future studies can overcome this limitation by collecting the data on the actual usage of e-learning from sources other than self-reported data, such as students' usage data from LMS. Finally, because our findings are based on a sample of students from Sri Lanka, their transferability to other countries with differing socioeconomic conditions may be limited. Future research can examine if the findings of this study hold true in various settings by replicating the empirical model employed in this study. Future studies can also incorporate other external variables into the model that could explain the variation in e-learning usage. Furthermore, research using different study designs, such as longitudinal studies, might produce more realistic results on the acceptance of e-learning among university students throughout the COVID-19 epidemic.

**Supplementary Materials:** The following are available online at https://www.mdpi.com/article/10.3390/educsci11080436/s1, Table S1: Sample of Questionnaire, Figure S1: Results of the PLS-SEM analysis, Figure S2: Bootstrapping Results of the PLS-SEM analysis, Figure S3: Results of the Moderating analysis.

**Author Contributions:** Conceptualization, P.R.W.; Methodology, P.R.W.; Formal Analysis, P.R.W., H.N.R. and W.H.M.S.S.; Data Curation, P.R.W., H.N.R., S.B.A. and W.H.M.S.S.; Writing—Original Draft Preparation, P.R.W., H.N.R., M.M.S.C.M., M.N., W.H.M.S.S.; Writing—Review and Editing, P.R.W., M.M.S.C.M., M.N., S.B.A. and W.H.M.S.S.; Supervision and Project Administration, M.N., S.B.A.; Funding Acquisition, P.R.W., H.N.R. and W.H.M.S.S. All authors have read and agreed to the published version of the manuscript.

**Funding:** The APC was funded by the AHEAD project (Grant no. AHEAD/RA2/ELTAELSE/RJT/ FMS/OVAA 26) of the Faculty of Management Studies, Rajarata University of Sri Lanka.

**Institutional Review Board Statement:** Not applicable.

**Informed Consent Statement:** Not applicable.

**Data Availability Statement:** The data presented in this study are openly available in Figshare at https://doi.org/10.6084/m9.figshare.14769468.v1 (accessed on 16 August 2021).

**Conflicts of Interest:** The authors declare no conflict of interest.

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
