# Peer review of "The COVID-19 Pandemic and the Acceptance of E-Learning among University Students: The Role of Precipitating Events"

_education, doi:10.3390/educsci11080436_

Round 1
Reviewer 1 Report
Not sure what e-mode is, needs clarification
E-education needs to be clearly defined.
What the authors understand by e-learning.
The statement page 4 "most universities around the world are using an e-leaning system to deliver their curriculum" can’t be verified unless the authors explain what they mean by e- learning
Majority of the HE Institutions use blended learning of hybrid learning
"The theoretical model developed in this study is basically derived from the TAM proposed by [10]" in here the authors name should be used ,According to ([23], p.181) ??? again the name authors name should be used this is repeated many time in the paper and should be changed
very often author refer to something as extensively and only refer to one reference source that is no enough, more reference should be used., "ample evidence" - and again only reference do one paper such situation needs to be crosscheck in the whole paper
a lot of unneeded words like basically are used, they should be removed or replaced
The hypothesis properly developed
Results properly benchmarked
Limitation of the study needs to be clearly stated
Possible extension/expansion of the research need to be added
Implications of the study and recommendation for students, academic and HE institution needed
Reviewer 2 Report
Authors verify the acceptance of e-learning technologies during Covid-19 pandemic. The Technology acceptance method has been used.
The article is well written and well organized. Experiments have been correctly conducted and explained. Massive experiments have been conducted, since more than 1000 subjects have been involved.
The quality of this article is very high.
Some minor remarks:
- Figures 1 and 2 are not clear. You should include a description of these images in the text. Moreover, please use high quality images
- I would suggest to use some graphs to summarize the content in table 2, since several tables make the reading difficult
